# Estimation of Oral Disease Burden among Older Adults in LTC: A Scoping Review

**DOI:** 10.3390/ijerph21030248

**Published:** 2024-02-21

**Authors:** Bathsheba Turton, Gheed Alqunaybit, Amrita Tembhe, Alaa Qari, Kadambari Rawal, Ernest Mandel, Joseph Calabrese, Michelle Henshaw

**Affiliations:** 1Henry M. Goldman School of Dental Medicine, Boston University, Boston, MA 02118, USA; galqu@bu.edu (G.A.); amritat@bu.edu (A.T.); kady@bu.edu (K.R.); jobean@bu.edu (J.C.); mhenshaw@bu.edu (M.H.); 2Faculty of Dentistry, Umm Al-Qura University, Makkah 02131, Saudi Arabia; ahqari@uqu.edu.sa; 3Hebrew Seniorlife, Harvard Medical School, Boston, MA 02131, USA; ernestmandel@hsl.harvard.edu

**Keywords:** long-term care, dental caries, tooth loss, periodontal disease, scoping review

## Abstract

Oral health is an essential part of healthy aging and very little data exists around the disease burden for older adults in a long-term care setting. The aim of this scoping review was to estimate the disease burden of dental caries, periodontal disease, and tooth loss among older adults in Long-Term Care (LTC). This scoping review was conducted in accordance with the Joanna Briggs Institute methodology. A detailed strategy was used to conduct a comprehensive search of electronic databases: PubMed, Embase, and Dentistry and Oral Sciences Source (DOSS). The Rayyan AI platform was used to screen abstracts for assessment by one of five co-investigators. Results indicate that only one in three might have a functional dentition upon entry into LTC, and among those who are dentate, most might expect to develop at least one new coronal and one new root caries lesion each year. There is a need to better document the disease experiences of this group to tailor approaches to care that might reduce the avoidable suffering as a result of dental caries and periodontal disease.

## 1. Introduction

Oral health is an essential part of healthy aging and oral disease among the aging global population, and has led to an estimated 28.02% increase in Disability Adjusted Life Years (DALY) for older adults between 1990 and 2010 [1]. Recent data from a nationally representative sample in the United States (US) suggest that one in six community dwelling older adults have untreated dental caries [2]. It is thought that those who live in Long-Term Care (LTC) facilities may have more severe experiences, although not a lot of data exist. Currently, in the US, 1.5 million older adults live in Long-Term Care (LTC). This number is increasing. LTC residents are largely female (68%), non-white Hispanic (75%), 65+ years old (84%), and Medicaid recipients (72%) [3]. It is estimated that their need for dental care is also growing, since almost 70% of older adults and 60% of nursing home residents are dentate.

The most prevalent oral diseases are dental caries and periodontal disease, which are both preventable Non-Communicable Diseases driven by socio-behavioral factors. In addition, these conditions are mediated by complex bacterial biofilm interactions. With regard to older adults, there are features, such as a shift in their diet towards refined carbohydrates, complex medical conditions, polypharmacy, and diminishing dexterity for performing oral hygiene practices, which contribute to increases in the rate of oral disease activity. In the caries process, the ecology of the dental biofilm system is driven by an individual’s diet and oral hygiene practices which, in turn, are driven by social, biological, and structural determinants [4].

Yet, the burden of both diseases remains persistent and progressive among marginalized communities such as those in LTC. This scoping review aimed to estimate the disease burden of dental caries (coronal and root carious lesions), periodontal disease, and tooth loss among older adults in LTC.

## 2. Materials and Methods

A scoping review was conducted using a methodology consistent with PRISMA_ScR guidelines for scoping reviews and their protocol. The protocol has been previously published in the open science framework (OSF) [5]. A detailed search strategy was developed to generate articles from three databases: PubMed, Embase, and Dentistry and Oral Sciences Source. The search terms for PubMed were built first and then adapted for the other search platforms:

PubMed Search terms: (((“Aged”[Mesh] OR “older adult*”[tw] OR “the elderly”[tw] OR “elderly adult*”[tw] OR “frail adults”[tw] OR elderly[tw] OR frailty[tw]) AND (“Mouth Diseases”[Mesh] OR “Tooth Diseases”[Mesh] OR “mouth disease*”[tw] OR “oral disease*”[tw] OR “periodontal disease*”[tw])) AND ((“long-term care”[mesh] OR (“long term”[tw] AND (care[tw] OR facilit*[tw] OR home*[tw] OR support[tw]))) OR (Mouthwashes[Mesh] OR “Cariogenic Agents”[Mesh] OR “Cariostatic Agents”[Mesh] OR Dentifrices[Mesh] OR mouthwash[tw] OR “mouth rinse”[tw] OR fluoride[tw] OR betadine OR “topical fluoride”[tw] OR “sodium fluoride”[tw] OR “tin fluoride”[tw] OR “phosphate fluoride”[tw] OR toothpaste OR “artificial saliva”[tw] OR “saliva substitute”[tw]))) NOT (“Infant”[Mesh] OR “Child”[Mesh] OR “Birth Cohort”[Mesh] OR “Young Adult”[Mesh] OR “Adolescent”[Mesh] OR child* OR “young adult” OR teenager OR teen OR kid OR baby OR infant) AND (1985:2022[pdat]).

In addition to structured searching, manual searching was used to identify any additional articles based on authorship or expert knowledge of the investigative team (6 out of 8 are gerontologists). The search terms have been previously published in OSF [5].

### 2.1. Study Selection

Abstracts were screened using the Ryyan platform [6] by the lead investigator for initial consideration in the scoping review. Then, full text versions of screened articles were reviewed by the investigative team for inclusion and data synthesis. The reason for exclusion and the stage at which the article was excluded were reported. Table 1 presents the inclusion and exclusion criteria.

### 2.2. Data Synthesis

Six calibrated investigators extracted data points using an electronic form (Microsoft Forms; Microsoft Corporation, Redmond, WA, USA). The form included sections on: paper characteristics, description of data points for relevant disease indicators, such as coronal caries, root caries, missing teeth, periodontal disease, and space for reviewers to comment on any study features such as limitations or unique features of study design. For each condition of interest, notes were made on the types of indices used and prevalence measures, severity, and incidence were recorded in the online form.

### 2.3. Calibration Process

A training video was developed by the first author, discussing the problems encountered by older adults, complexity of the problem, the objectives of the review, and the extraction process using the form. Other videos discussed the definitions of oral diseases and a standard way of extracting them from papers, and discussed the initial review conducted by the team for calibration. Each investigator reviewed 10 papers as part of a pair and when there was sufficient consistency between the pair, the investigator evaluated papers independently (A total of 50 of 172 papers were reviewed by two investigators).

### 2.4. Synthesis of Results

Summary statistics were prepared to describe the types of studies included and among which populations. Following this, data on the disease burden and disease incidence were collated and summarized. Data on means and standard deviations were aggregated, and then the Cochrane formula was used to provide a summary measure wherever possible [7]. Increment or incidence data were presented, then values were standardized to indicate the increment or incidence per year. Where data from both the community and Long-Term Care facilities were presented, only the data from the LTC setting were presented in the summary tables.

## 3. Results

Figure 1 presents data on the papers included in the scoping review. Twenty-nine papers had data on tooth loss, 26 on coronal dental caries, 26 on root caries, and 21 on periodontal disease.

Data presented on missing teeth suggest that most older adults in Long-Term Careare missing more than 10 teeth. Two papers, one out of India [8] and another out of Sweeden [9], found older adults in LTC had a mean of less than ten teeth missing (Table 2). Overall, approximately one in three were edentulous, and one in three might have a functional dentition (more than 20 teeth). Data on the incidence of tooth loss among older adults in Long-Term Care was not present. Papers using community-dwelling samples suggest that between 0.1 and 0.5 teeth per person per year might be lost among 5–10% of the population [10,11,12,13,14,15].

Data suggest that the mean number of untreated coronal caries lesions present among older adults in LTC was between one and four teeth (Table 3). The mean number of Decayed Missing and Filled Teeth (DMFT) in most papers exceeded 20 teeth. The incidence of new lesions was around two teeth per year, and one paper suggested that 17.7% of eligible surfaces might develop carious lesions (attack rate).

Data on root carious lesions were presented using a variety of different indices: the root caries index, Nyvad modified criteria, Fejerskov et al.’s criteria, and ICDAS II (Table 4). There was a large variation in the number of teeth with untreated root carious lesions. Around 2/3 of the individuals in each sample had one or more untreated lesion(s), and most papers suggested that more than two teeth per individual might be involved. Data on disease progression suggests that between 10% and 20% of root lesions might progress yearly. In addition, data suggest that older adults in LTC might develop at least one new root carious lesion per year.

Most papers presented data on worst pocket depth, most severe Community Periodontal Index of Treatment Need (CPITN) score, or worst attachment loss (Table 5). When all data are considered, around two in five might be expected to have one or more teeth with pocket depths in the 4–6 mm range (equivalent to CPITN code 3) or worse, and one in five might be expected to have one or more pockets that are deeper than 6 mm. Community-based data suggest that one in five pockets progress yearly among one in ten individuals [13,55,56].

## 4. Discussion

This scoping review aimed to synthesize data from the existing literature to better understand the disease burden among older adults in LTC settings. There was a lack of consistency in reporting root caries and periodontal disease, making results among papers challenging to compare. The results suggest that over two-thirds of older adults present in LTC without a functional dentition and with many active and untreated coronal and root carious lesions. Data on periodontal conditions are less clear due to heterogenicity in data reported; potentially, one in five older adults in LTC might benefit from intensive periodontal treatments to manage deep periodontal pockets. These findings are consistent with global reports that suggest that older adults are entering the later stages of life with some teeth remaining [67].

Accessing robust epidemiological data for older adults in LTC is challenging, and this scoping review marks the beginning of a program of work to redesign oral health care for older adults in LTC in a US context. In addition, representative sampling is challenging due to difficulties ranging from logistical limitations associated with undergoing detailed dental examinations to gaining consent from those who might be cognitively impaired. These limitations mean that those with a more severe disease experience may be more likely to be excluded from sampling, and as such, reports may underestimate the disease burden among this population.

This scoping review had limitations where the search terms may have missed some papers with relevant data. Some of those relevant papers were picked up by manual searching. However, valuable datasets may likely have been excluded from consideration. While this scoping review does not precisely estimate disease experience, it does confirm that the disease experience is more severe than that reported for community-dwelling older adults [68].

The synthesized data suggest some consistencies in the cross-sectional pattern of disease whereby around 1/3 appear to be edentulous, 1/3 with less than 20 teeth (but not edentulous), and 1/3 with a functional dentition (>21 teeth). It seems that untreated caries lesions (coronal or root surfaces) are not present among all dentate residents of LTC but the distribution is not well defined, except in the NZ study where 2/3 of dentate adults had untreated lesions on coronal portions of teeth and 1/3 had untreated lesions on root caries [28]. Other studies with comparable data appear to share similar estimates [23,34,40,53]. While these patterns are described at a cross-sectional level, the disease progression is less well understood. For example, there are estimates of the mean number of new carious lesions. Still, there does not appear to be consistent data on the proportion of the population with disease progression while in LTC. Understanding the profile of disease incidence at a population level could help better design more efficient care pathways, thereby increasing access to care for the proportion of the population that most needs active intervention.

The current care pathway is inadequate for delivering dental care for LTC residents. The barriers to treatment for older adults, particularly those in LTC, include difficulty accessing transportation to the dental office and relying on caregivers to recognize their need for routine care [69]. The ideal approach would be to address oral disease at earlier stages of disease and while individuals possess greater levels of independence. In the context of LTC, the dental care delivery system is not well-positioned to address the needs of older adults. The workforce has few gerontologists, older adults often lack dental insurance, and the payment structure in the private practice model does not account for older adults who have multifaceted medical histories, take longer to treat, and have complex clinical manifestations. A preventive approach would provide an opportunity to shift away from the presently nonviable dentist-centric approach and reduce the need for unexpected surgical interventions (dental emergency) that can disrupt the lives of those in LTC.

Designing a care pathway that caters to the needs of the older adults in the LTC population requires consistent and reliable data and along with that robust health surveillance systems. Dental caries and periodontal disease conditions are not equitably distributed across the population; rather they exhibit a skewed profile where most disease occurs among a subgroup of individuals. What is unclear from the data is what proportion are likely to have active caries or have active periodontal disease. This highlights the need to document better the natural history of oral disease and its distribution among older adults in LTC. Further, the findings of this scoping review highlight the need to document disease experience using standardized approaches to recording data, particularly regarding root caries and periodontal disease, on which very few comparable studies exist. One example of this type of contention is around the definition of ‘periodontal disease’ in this population who are likely to have both gingival pockets and attachment loss. Studies only looking at combined attachment loss (distance from the cementoenamel junction to the bottom of the pocket) are likely to overestimate the presence of active periodontal disease [65,66] as a proportion of the attachment loss might be due to recession that might be an expected part of normal aging rather than an actively inflamed gingival pocket [70]. By contrast, using only indicator teeth to screen for the presence of bony defects in periodontal supporting tissues using the CPITN may underestimate the level of disease present [28].

While there are efforts to standardize data collection for older adults when they enter LTC, [70] these screening tools have yet to be linked to care plans that might predictably maintain the highest attainable state of oral health. Such an understanding might help to inform a population-level approach to preventing suffering that occurs as a result of preventable oral disease.

Recent literature on interventions has heavily focused on plaque removal and on training caregiving staff to perform better oral hygiene care or undertake dental screening procedures [71,72]. However, there are varying results due to the challenges of managing long-term care facilities where staff turnover is high and the demands on caregiver time are great [73]. Very few interventions have demonstrated disease prevention regarding caries increment or management of periodontal pocket progression. Managing biofilm-mediated diseases from a preventive paradigm in LTC settings is complex because the conditions are juxtaposed against the unique social and functional structure of institutionalized living, where specific power dynamics among staff and between staff and residents make it difficult to address the socio-behavioral drivers of the disease [74]. It is well documented that when an individual loses autonomy and has a diminished ability to set personal goals, then health outcomes are compromised, including increases in oral disease [75]. This is especially true for LTC residents, where increased frailty and dependence can trigger declining oral health [76]. Therefore, there is a tremendous need for novel and sustainable oral health interventions in LTC.

## 5. Conclusions

This scoping review brings aggregated reported data on the oral health of older adults in long-term care. There appears to be some consistency in the present disease patterns. Only one in three might have a functional dentition upon entry into LTC, and among those who are dentate, most of them might expect to develop at least one new coronal and one new root caries lesion each year. There is a need to document the disease experiences of this group better to tailor approaches to care that might reduce the avoidable suffering as a result of dental caries and periodontal disease.

## Figures and Tables

**Figure 1 ijerph-21-00248-f001:**
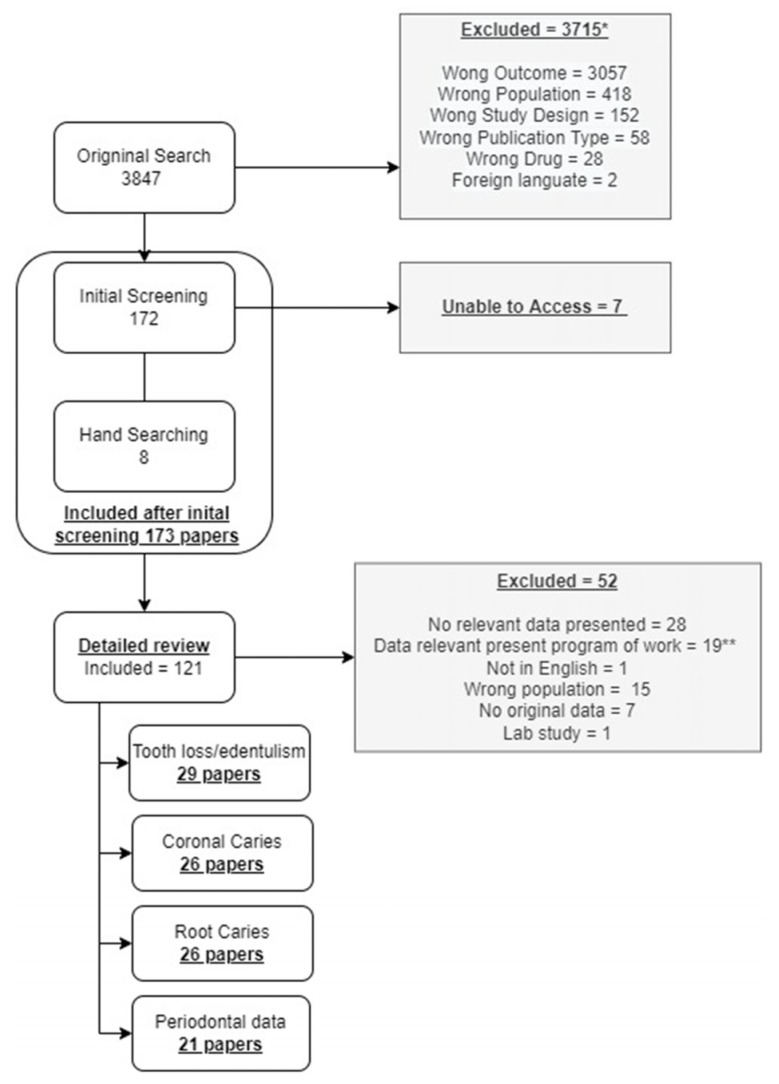
Description of included papers in the scoping review. Some papers were included in more than one disease category, while others inform alternative study questions outside the scope of the present paper. * Papers may have been excluded for more than one reason. ** 19 papers were included in the broader scoping review but not included in the present report. Underlined text represents subtotal counts.

**Table 1 ijerph-21-00248-t001:** Criteria for inclusion and exclusion of papers.

	Inclusion Criteria	Exclusion Criteria
Population	Older adults in long-term care facilities and community settings and in any country.	Younger population age groups (<65 years old)
Language	English	Non-English language papers
Interventions	Those include, but are not limited to, Fluoride Varnish, Mouth rinses, Silver Diamine fluoride, high fluoride toothpaste	Interventions occurring with the wrong population.
Publication status	Published literature searchable by PubMed, Embase, or Dentistry and oral sciences sources.	Not available online using search engines or hand searches
Study designs	Cross-sectional surveys, cohort studies, experimental studies	Qualitative studies, review articles, articles without original data
Outcome measures	Caries incidence, incidence of tooth loss (incidence), periodontal disease incidence, oral symptoms, periodontal diseases	If there is an absence of clinical measures on caries, periodontal disease, tooth loss, or oral symptoms. Studies where the clinical outcome is the plaque score and there are no other clinical outcome measures meeting our criteria will be excluded.
Dates	Any date if the electronic version of the paper is available	All dates included

**Table 2 ijerph-21-00248-t002:** Tooth loss and edentulism among older adults in long term care.

Author	Country	Prevalence of Edentulism	Mean Number of Missing Teeth	Proportion with >20 Teeth	Incidence of Tooth Loss
Stuck et al., 1989 [16]	Switzerland	58.90%			
Hunt et al., 1995 [10] *	USA				0.7 (SD2.0) over 18 m; 0.5/year
Chalmers et al., 2002 [17]	Australia	66%	18.9 (no SD)		
Chalmers et al., 2002 [18]	Australia		19.3 (SD 7.4)		
Chalmers et al., 2002 [19]	Australia	excluded	10.4 (No SD)		0.4/year
Fure, 2003 [20] *	Sweden	27%	21.2 (SD 6.6)		2.5 (SD 3.4) over 5 years; 0.5/year
Wyatt & MacEntee, 2004 [21]	Canada		14.9 (SD 6.9)		0.1 teeth per year
Nevalainen et al., 2004 [11] *	Finland	31%	16.2 (SD 8.0)		1% year became edentulous
Ajwani & Ainamo, 2004 [12] *	Finland		12.2 (SD 7.7)		0.9 teeth over 5 years; 0.2 teeth per year
Chalmers et al., 2005 [22]	Australia	63.10%			
Adam et al., 2006 [23]	UK	65.90%	27.8 (SD 7.0)		
Tramini et al., 2007 [24]	France	26.90%		33.6%	
Arpin S et al., 2008 [25]	Canada		19.1 (SD 7.8)		
Yoshihara et al., 2008 [13] *	Japan		19.3 (SD 8.6)		1.7 (SD 2.2) over 6 years; 0.3 teeth lost per year
Philip P et al., 2012 [26]	Australia		17.7 (SD 7.2)		
Zenthöfer et al., 2014 [27]	Germany	39.40%	20.3 (SD 8.9)		
Agrawal et al., 2015 [8]	India	37.90%	6.7 (SD 5.2)		
Tompson et. al., 2015 [28]	NZ	56.6%	15.6 (SD 14.7) **	35.1% **	
Matsuyama et al., 2016 [14] *	Japan	8%			8.2% lost ≥ 1 tooth over 3 years (2.7%/year)
Gülcan et al., 2017 [15] *	Norway and Sweden				5.5% lost one or more teeth over 5 years (1.1%/year)
Pham and Nguyen, 2018 [29]	Vietnam	9.90%	13.8 (SD 7.4) **	52.1%	
Saarela et al., 2019 [30]	Finland	13% ***			
Chiesi and Grazzini et al., 2019 [31]	Italy	39.8%		21%	
Girestam Croonquist et al., 2019 [32]	Sweden		7.8 (SD 3.0) **		
Tanji et al., 2020 [33]	Japan	32% <10 teeth		44.4%	
Bianco A et al., 2021 [34]	Italy	31.90%		17%	
Ericson et al., 2022 [9]	Sweden		8.0 (SD 5.9) **		
Tokumoto et al., 2022 [35]	Japan		14.7 (SD 8.0)		

SD = Standard Deviation * Data are from a community setting rather than a Long-Term Care setting ** only dentate individuals were included *** Edentulous without dentures.

**Table 3 ijerph-21-00248-t003:** Coronal dental caries among long term care residents.

Author	Country	Caries Severity	Untreated Caries Severity and Prevalence	Caries Increment/Prevalence of New Lesions (DS/DT)
MacEntee et. al., 1985 [36]	Canada		78%	
Vigild et al., [37]	Denmark		DT = 9.6	
Jones et al., 1993 [38]	USA	FS = 9.5 (SD 5.7)	DT = 1.0 (SD 1.5)	
MacEntee et al., 1993 [39]	Canada	DMFT = 24.1	DT = 3.4 (SD 4.4) DS = 7.4 (SD 12.1)	DS/year = 0.9 (SD 1.5) FS/year = 0.5 (SD 1.3)
Wyatt, 2002 [40]	Canada	DMFS = 112.3 (SD 26.6)	DS = 3.8 (SD 4.2)	
Chalmers et al., 2002 [17]	Australia	DMFT = 23.7		
Chalmers et al., 2002 [18]	Australia	DMFT = 24.4 (SD 4.5)	DT = 1.7 (SD 2.5)	17.7% (SE11.7%) **
McMillan et al., 2003 [41]	Hong Kong	DMFT = 21.4 (SD 0.6)	DT = 2.1 (SD 0.2)	
Wyatt and MacEntee., 2004 [21]	Canada	DMFS = 112.3 (SD 26.6)		DT/year = 2.3 (SD not presented)
Chalmers et al., 2005 [22]	Australia			DT/year 2.2 (SD 3.9)/year
Adam et al., 2006 [23]	UK	DMFT = 29.7 (SD 11.7)	DT = 0.9	
Vilstrup et al., 2007 [42]	Denmark	FS = 11.8	DS = 3.0 (SD 3.4); 57.60%	
Arpin et al., 2008 [25]	Canada	DMFT = 24.9		
Gluhak et al., 2010 [43]	Austria	DMFT = 25.6 (SD 4.2)		
Chen et al., 2010 [44]	United States	DMFT = 23.2	DT = 3.3 (SD 4.0)	
Ellefsen et al., 2012 [45]	Denmark	FS = 32.4 (SD 21.4)	DS = 2.9 (SD 3.6)	
Philip P et al., 2012 [26]	Australia	DMFT = 26.0 (SD 4.3) *	DT = 3.0 (SD 3.5)	
Silva et al., 2014 [46]	Australia	DMFT = 21.7 (SD 0.3)	DT = 2.7 (SD 0.2)	
Bilder et al., 2014 [47]	Israel		DT = 4.2 (SD 4.5)	
Agrawal et al., 2015 [8]	India	DFT—8.3 *	DT = 1.5 (no SD reported)	
Tompson et. al., 2015 [28]	NZ	DMFT = 24.2 (CI 23.5, 25.0)	DT = 2.2 (CI 1.8, 2.5) 61.3%; DT among those with caries = 3.5 (CI 3.1, 4.0)	
Holtzman et al., 2015 [48]	USA		DT = 0.50 (26.0%)	
Pham & Nguyen, 2018 [29]	Vietnam	DMFT = 20.0	DT = 5.8 (SD 4.0)	
Zhang et al., 2019 [49]	Hongkong, China	DMFT = 10.1 (SD not reported)	DT = 0.8 (SD 1.6)	
Bianco et al., 2021 [34]	Italy	DMFT = 26.4 (SD 7.5)	DT—3.5 (SD 4.6); 70.8%	
Ericson et al., 2022 [9]	Sweden		DT—2.0 (SD 6.7)	

DMFT/DMFS refer to the Decayed Missing and Filled index where data are presented at Teeth (T) level or Surface (S) Level * Value explicitly excludes edentulous from the sample; ** Caries attack (% of eligible surfaces to become carious).

**Table 4 ijerph-21-00248-t004:** Root carious lesions among long-term care residents.

Citation	Country	Root Caries Definition **	Severity/Prevalence of Root Caries **	Incidence and Increment ***
Jones et al., 1993 [38]	USA	NIDCR definition	T-DRS = 2.8 (SD 4.3) T-DRS = 4.6 (SD 4.7)	T-DRS/year = 1.0 (SD 3.3)
MacEntee et al., 1993 [39]	Canada	NS		T-DRS/year = 1.6 (SD 3.1) Filled root surfaces = 0.8 (SD 2.3)/year
Berg et al., 2000 [50]	USA	NIDCR diagnostic criteria.	DRS = 4.1 (SD 8.9) FRS = 21.1 (SD 18.5)	
Wyatt et al., 2012 [40]	Canada	The Root Caries Index	DFRS = 30.3 (SD 26.1)	
Chalmers et al., 2002 [17]	Australia	The Root Caries Index	T-DRS = 1.5; T-FRS = 1.1	
Chalmers et al., 2002 [18]	Australia	The Root Caries Index	T-DRS = 2.6 (SD 3.9) T-DFRS = 1.3 (SD 3.3)	
McMillan et al., 2003 [41]	Hong Kong	NS	T-DRS = 1.3 (SD 0.2) teeth	
Wyatt and MacEntee, 2004 [21]	Canada	The Root Caries Index	T-DRS= 4.6 (SD 6.8); 68% of people	1.8 (SD 1.6)
Chalmers et al., 2005 [22]	Australia	NIDCR definition		0.8/year
Tan et al., 2010 [51]	Hongkong	Lesions “easily penetrable with a sharp sickle probe”	DRS = 1.3 (SD 0.1); FRS = 0.8 (SD 0.1) 10.7% of exposed root surfaces	1.3 (SD 0.2) or 0.4/year
Ellefsen et al., 2012 [45]	Denmark	NIDCR criteria	DRS = 4.9 (SD 6.1); FRS = 5.4 (SD 4.8)	
Philip P et al., 2012 [26]	Australia	The Root Caries Index	0.1 (SD 0.4)	
Ekstrandet al., 2013 [52]	Denmark	NS	3.3 (SD 3.0) *	
Zhang et al., 2013 [49]	China	NS	T-DRS = 0.2 (0.1)	1.0 (SD 0.1)/year
Silva et al., 2014 [46]	Australia	ICDAS II ≥ Code 2	T-DRS = 3.4 (0.3)	
Tompson et al., 2015 [28]	NZ	A lesion on the root surface that was soft to exploration using a periodontal probe	T-DRS = 0.8 (CI 0.6, 1.0); 33.7%—among those with >1 lesion TDRS = 2.4 (CI 2.0, 2.7)	
Pham and Nguyen, 2018 [29]	Vietnam	NS	T-DRS = 6.0 (SD 4.2)	
Zhang et al., 2020 [53]	Hongkong, China	Visual inspection (NS)	T- DR = 0.7 (±1.7); T-DFS = 1.3 (SD 2.1); 43.10% of people	
Patel et al., 2022 [54]	UK	NS	T-DFRS = 3.1; 69.20% of people	
Ericson et al., 2022 [9]	Sweden	Nyvad modified criteria		11.4% of lesions progressed
Tokumoto et al., 2022 [35]	Japan	Fejerskov et al.	16% of teeth	T-DRS = 14.6% of teeth/year and 22.5% of existing lesions progressed across one-year

NIDCR = National Institute for Dental and Craniofacial Research; NS = Not stated; T-DRS = Tooth level indicator for one of more Decayed Root Surfaces; DRS = Decayed root surface as a surface level indicator; FRS = Filled root surface as a surface level indicator; T-DFRS = Tooth level indicator for the Decay or Filling on one or more root surface; T-DFS. * Index used to classify carious lesions on the root surface ** Severity is defined in terms of the number of lesions where as prevalence is defined as the proportion of individuals. *** Incidence is the proportion of individuals with one or more new lesions and increment is the number of new carious lesions on root surfaces per year.

**Table 5 ijerph-21-00248-t005:** Periodontal disease among LTC residents.

Author	Country	Disease Descriptor	Perio Status/Incidence of Disease
MacEntee et al., 1985 [36]	Canada	CPITN	CPI > 3 = 29%
Beck et al., 1995 [55]	USA **	Incidence = % of people with 1+ sites of AL of 3+ mm over 3 years Affected = mean number of sites with AL in people with AL.	One or more new sites of attachment loss = 27.5% of people. Progression of existing sites with attachment loss = 11.1% of people Overall (either new or progressing sites) = 20.1% The mean number of sites progressing = 4.5 sites (SD 0.5)
Ogawa H et al., 2002 [57]	Japan *	Worst CAL	<4 mm = 3.5%; 4–6 mm = 32.5%; >6 mm = 64%
Levy et al., 2003 [58]	USA **	Worst attachment loss	For those who had their most severe CAL at 4 mm, the mean number of sites involved was 11.38 (±12.47); 6 mm = 1.93 ± 5.05; 8 mm = 0.44 (1.62)
McMillan et al., 2003 [41]	Hong Kong **	The most severe CPITN score	CPI 0 = 0.0%; CPI 1 = 0.5%; CPI 2 = 40.1%; CPI 3 = 39.6%; CPI 4 = 19.8%
Mack et al., 2004 [59]	Germany *	Presence of pockets >4 mm and >6 mm	≥4 mm pockets = 49.1%; ≥6 mm pockets = 21.7%
Ajwani and Ainamo, 2004 [12]	Finland **	One or more sextants with > Code 3 CPITN	CPI ≥ 3 = 43%
Qian et al., 2007 [56]	USA *	CAL, Incidence of ALOSS (>2 mm difference).	Mean CAL = 1.6 mm (SD 0.6); 16.1% of sites with >2 mm progression/10 years.
Yoshihara et al., 2008 [13]	Japan *	Presence of site with >3 mm difference in periodontal readings across one-year	Sites per year = 9.8 (SD 6.5)
Orwoll et al., 2009 [60]	USA *	Gingival index, Gingival bleeding, CAL	Mean gingival index = 1.2 + 0.5; Prevalence of gingival bleeding = 53%; Mean CAL = 3.0 (SD 0.8)
Chen et al., 2010 [44]	USA *	Presence of calculus, plaque and gingival bleeding;	none= 1.2%; mild to moderate = 81.3%; High = 17.7%
Gluhak et al., 2010 [43]	Austria	CPITN	CPI ≥ 2 = 84.1%
Siukosaari et al., 2010 [61]	Finland **	Most severe CPITN	CPI ≥ 3 = 44.7
Sánchez-García et al., 2011 [62]	Mexico *	% of those with CPI code >2	CPI > 2 = 36.1%
Syrjälä et al., 2012 [63]	Finland *	Number of teeth with pocket depth >4 mm	Mean of 2.9 teeth (SD3.7)
Gaio et al., 2012 [64]	Brazil *	Worst CAL (person level)	W-CAL ≥ 3 mm = 94%; W-CAL ≥ 5 mm = 60% of teeth
Tompson et al., 2015 [28]	NZ	CPI code on index teeth	CPI ≥ 3 = 11.2% (CI 7.9, 14.5) CPI > 4 = 2.1%
Agrawal et al., 2015 [8]	India	CPITN by sextant	0.5 sextants/person (SD 0.9);
Pham and Nguyen, 2018 [29]	Vietnam	Worst pocket depth	≤3 mm = 526 (73.8%); 4–6 mm = 116 (16.7%); ≥7 mm = 68 (9.5%)
Kimble et al., 2022 [65]	US/UK *	Percentage people with 20% of sites having >3.5 mm CAL	UK sample—>20% with >3.5 mm pockets = 20.1% US sample—>20% with >3.5 mm pockets = 52.9%
Kotronia et al., 2022 [66]	UK/US *	>20% sites with LOA >3.5 mm and >5.5 mm	UK sample—20% >3.5 mm = 53%; 20% > 5.5 mm = 29% US sample—>20% >3.5 mm = 64%; >20% > 5.5 mm = 31%

CPI = Community Periodontal Index of Treatment Need; CAL = Combined Attachment Loss; LOA = Loss of Attachment * community setting; ALOSS = Attachment loss ** mixed community and long term care setting.

## Data Availability

Further data supporting reported results can be found at https://doi.org/10.17605/OSF.IO/6FVE5; accessed on 19 February 2023.

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
