# Peer review of "Estimation of Oral Disease Burden among Older Adults in LTC: A Scoping Review"

_ijerph, 2024, doi:10.3390/ijerph21030248_

Round 1

Reviewer 1 Report

Comments and Suggestions for Authors

The main question of this scoping review was to estimate the burden of dental caries, periodontal disease, and tooth loss among older adults in Long Term Care (LTC). It is an important question  and this frail group need to get attention regarding their oral health. Both dental professionals and healthcare professionals should be mindful of the problems that may exist, but oral care and treatment of oral diseases is unfortunately often a neglected area in LTC. Compiling knowledge as this review has done is difficult as the included studies often differ in terms of methodology such as assessment instruments and diagnosis criteria. It is a well-written paper, but I have a few points of view to consider.

 Abstract, line 12: The first time LTC is mentioned, the abbreviation should be written in parentheses.

Material and methods, page 2, line 52: You have done an ambitious search using many search terms and I agree with your discussion that you may have missed terms. Dental caries, tooth decay, tooth loss and edentulous could have been relevant to your purpose. 

Results, Figure 1, page 4: There is a question mark regarding the number of included items (n=172). Original search gave 3847 publications and excluded publications were 3715= 132. It is also unclear what is meant by the character * after 3715 (excluded publications).

Page 4, line 114. Please insert which references included community dwelling long term care.

Page 5, Table 3. See Wyatt 2002 under the heading "Untreated caries severity and caries prevalence". It states 3.8. What does this mean?

Discussion, page 8, line 158: Do the results indicate that two-thirds of the elderly enter LTC without functional teeth and with many active and untreated coronal and root lesions? Most likely, there are differences in how long the older ones have had LTC, some have recently entered LCT and others have had it for a longer period. 

Additional comments: You had no limit on the year of publication. The included papers are published between 1985 and 2022, nearly forty years. Do you think that the situation is same among the including participants during this long time period?

You discuss the difficulties delivering dental care to this group in focus. There are probably also difficulties to examine the dental status, which affects the results (over- or underestimated diagnosis), that also need attention in the discussion.

Author Response

Thank you for taking the time to review our manuscript. I have numbered off your feedback points and placed our responses in italics below. In addition, changes within the manuscript are presented in yellow highlights.

  1. Abstract, line 12: The first time LTC is mentioned, the abbreviation should be written in parentheses.

Correction made as suggested

  1. Material and methods, page 2, line 52: You have done an ambitious search using many search terms and I agree with your discussion that you may have missed terms. Dental caries, tooth decay, tooth loss and edentulous could have been relevant to your purpose.

Thank you for the chance to clarify these points. In the search the syntax indicated certain terms which were catalogued ‘Mesh’ terms meaning that the search encompasses a broad subsection of key words under each Mesh Category. For example, the term ‘Mouth Diseases’ is a branch of ‘Stomatognathic Diseases’. The Mouth Diseases term includes 33 branches and additional sub-branches. We cast our search wide to capture as many oral health related pieces of work and then later filtered more stringently according to the population and setting of interest. The term ‘Tooth Diseases’ has 21 branches and numerous sub branches from those. Between these two mesh terms all terms that the reviewer has listed were covered. 

  1. Results, Figure 1, page 4: There is a question mark regarding the number of included items (n=172). Original search gave 3847 publications and excluded publications were 3715= 132. It is also unclear what is meant by the character * after 3715 (excluded publications).

Thank you for picking this up. Upon examination, the ‘*’ and ‘**’ subscript descriptions were missing from the version of the paper sent out for review. These descriptions have now been replaced.

*Papers may have been excluded for more than one reason

**19 papers were included in the broader scoping review but not included in the present report

  1. Page 4, line 114. Please insert which references included community dwelling long-term care.

Relevant references have been denoted in the text and also using the * superscript in the relevant table.

“Papers using community-dwelling samples suggest that between 0.1 and 0.5 teeth per person per year might be lost among 5–10% of the population [11, 17, 18, 23, 27, 28].”

  1. Page 5, Table 3. See Wyatt 2002 under the heading "Untreated caries severity and caries prevalence". It states 3.8. What does this mean?

Thank you for pointing out this omission. The abbreviation ‘DS’ was added. The definition of that abbreviation is spelled out in the subtext of the table as ‘Decayed Surfaces’

  1. Discussion, page 8, line 158: Do the results indicate that two-thirds of the elderly enter LTC without functional teeth and with many active and untreated coronal and root lesions? Most likely, there are differences in how long the older ones have had LTC, some have recently entered LCT and others have had it for a longer period.

We agree with the point that the reviewer is making; that there are likely to be differences in disease severity based on the length of stay of an individual within the LTC facility. The scoping review doesn’t allow us to make the differentiation of disease severity by length of stay. For this reason the sentence has been updated as follows:

“The results suggest that over two-thirds of older adults present in LTC without a functional dentition and with many active and untreated coronal and root dental caries lesions.”

  1. Additional comments: You had no limit on the year of publication. The included papers are published between 1985 and 2022, nearly forty years. Do you think that the situation is same among the including participants during this long time period?

It is clear that the manifestation of oral disease at a global level has changed drastically over 40 years. Specifically the prevalence of edentulism has decreased significantly especially among high income nations. The dominant narrative is that older adults are retaining their teeth longer and this drives a greater need for restorative care among said group.

In this scoping review, we see insufficient consistency in the sampling strategies or disease outcome measures within to assess those trends over time. In saying that, it is interesting to note that once data from the wealthy West are considered on their own (and without contributions from the community setting), then there are no clear, time-dependent trends in the prevalence of edentulism or mean DMFT values. One of the unique features of individuals entering LTC, particularly among high-income Western nations, is that those individuals are not representative of their underlying population. They have certain sets of features that set them apart from those who are ‘ageing in place’ at their homes or have not reached an advanced age where they seek assisted living. These features could be contributing to the consistency of oral disease in this unique setting over time.

Summary of DMFT data from wealthy west

Author

Country

Caries Severity

MacEntee et al., 1993 [39]

Canada

DMFT = 24.1

Chalmers et al., 2002a [12]

Australia

DMFT = 23.7

Chalmers et al., 2002b [13]

Australia

DMFT = 24.4 (SD 4.5)

Adam et al., 2006 [20]

UK

DMFT = 29.7 (SD 11.7)

Arpin et al., 2008 [43]

Canada

DMFT = 24.9

Gluhak et al., 2010 [44]

Austria

DMFT = 25.6 (SD 4.2)

Chen et al., 2010 [45]

United States

DMFT = 23.2

Wyatt and MacEntee, 2012 [24]

Australia

DMFT = 26.0 (SD 4.3) *

Silva et al., 2014 [47]

Australia

DMFT = 21.7 (SD 0.3)

Tompson et. al., 2015 [26]

NZ

DMFT = 24.2 (CI 23.5, 25.0)

Bianco et al., 2021 [34]

Italy

DMFT = 26.4 (SD 7.5)

Summary of edentulism prevalence in the wealthy west

Author

Country

Prevalence of Edentulism

Chalmers et al., 2002a [12]

Australia

66%

Chalmers et al., 2005 [19]

Australia

63.10%

Tramini et al., 2007 [21]

France

26.90%

Zenthöfer et al., 2014 [25]

Germany

39.40%

Chiesi & Grazzini et al., 2019 [31]

Italy

39.80%

Bianco A et al., 2021 [34]

Italy

31.90%

Tompson et. al., 2015 [26]

NZ

56.60%

Stuck et al., 1989 [10]

Switzerland

58.90%

Adam et al., 2006 [20]

UK

65.90%

  1. You discuss the difficulties delivering dental care to this group in focus. There are probably also difficulties in examining the dental status, which affects the results (over- or underestimated diagnosis), that also needs attention in the discussion.

We agree, and this point is acknowledged: Discussion, Para 2, sentence 1 – " Accessing robust epidemiological data for older adults in LTC is challenging.”

Regarding periodontal disease and root lesions, we highlight the differences in indices used within the table by defining the various indices used. We also avoid making any comments about trends over time. We modified the discussion to further emphasise the variation in disease measures.

“There was a lack of consistency in reporting root caries and periodontal disease, making results among papers challenging to compare.”

“Data on periodontal conditions are less clear due to heterogeneity in data reported.”

Regarding measurements of edentulism, tooth loss, and DMFT – these indices have been field tested for over 100 years in dental epidemiology and are proven to be ‘reliable’ measures of disease even if they don’t capture the full spectrum of manifestations of the caries process. We aimed to present data on both DMFT and DMFS where they were available, and the presence of these two adjacent disease measures can create ambiguity when comparing papers; further, DMFS may be less reliable than DMFT data.

Regarding disease incidence and progression measures, there is a major gap in our understanding of disease, which we point to in the following sentence.

“While these patterns are described at a cross-sectional level, the disease progression is poorly understood. For example, there are estimates of the mean number of new carious lesions.”

Reviewer 2 Report

Comments and Suggestions for Authors

line 31 - reformat the sentence -

Currently, in the US,1.5 million older adults live in Long Term Care (LTC). and that This number is increasing.

line 33 - It is estimated that their need for....

line 37-38 - Non-Communicable Diseases driven by socio-behavioral factors, and mediated by complex biofilm interactions. Does not make sense. sees like for verbiage is missing.

line 41-43  This scoping review aimed to estimate the disease burden of dental caries (coronal and root carious lesions), periodontal disease, and tooth loss among older adults in LTC. using a scoping
review approach.

line 46 - The  A scoping review

line 109-111 - Data presented on missing teeth suggest that most older adults in long-term care are missing more than 10 teeth. with just Two papers, one out of India [8] and another out of Sweeden [9], where found older adults in long-term care had a mean of less than ten teeth missing ....

Sentence needs to be written clearer.

line 159-160 -  ...coronal and root dental carious lesions.

line 174 - ...treated caries carious ... 

line 201 - ... proportion are likely to be caries active to have active caries  and or have active periodontal disease.

line 236-237 - This scoping review brings aggregated reported data on the oral health of older adults in long-term care. and There appears to be some consistency in the present disease patterns.

Comments on the Quality of English Language

As noted, I found a few grammatical issues. Other wise, the paper has good merit.

Author Response

Thank you for taking the time to review our manuscript. I have numbered off your feedback points and placed our responses in italics below. In addition, changes within the manuscript are present in yellow highlight.

We thank the reviewer for their support in adjusting our grammatical errors. Considering line numbers at the time of review - Suggested adjustments have been made to the following sections:  Line 31, Line 33, line 41-43, line 46, line 109-111, line 159-160, line 174, line 201, line 236-237 (; line numbers have changed in the present version and so changes have been highlighted in the manuscript using yellow highlight.

line 37-38 - Non-Communicable Diseases driven by socio-behavioural factors, and mediated by complex biofilm interactions. Does not make sense. sees like for verbiage is missing.

Sentences were re-written as follows: “The most prevalent oral diseases are dental caries and periodontal disease, which are both preventable Non-Communicable Diseases driven by socio-behavioural factors. In addition, these conditions are mediated by complex bacterial biofilm interactions.”

We also note that the reviewer picked up on our inconsistencies with regards to terminology around root caries and ‘root carious lesions’. We use the search function in word to identify further instances where the term ‘root caries lesions’ had been incorrectly used.

Reviewer 3 Report

Comments and Suggestions for Authors

Thank you for the opportunity to review your manuscript, entitled "Estimation of Oral Disease Burden among Older Adults in LTC: A Scoping Review".  It was a pleasure to read a well-written manuscript with sound and well-presented methodology and results.

While there are no significant concerns for me to raise, I would invite your team to consider the following points that I feel may improve your manuscript:

1. Introduction: While I think that the introduction sets a good background to the issue of interest, the caries process is oversimplified particularly within the context of older adults in LTC.  Only biofilm and the influence of diet are discussed, even though these would not necessarily be any different from the rest of the population.  A more detailed discussion about other significant risk factors in this population that may place them at higher risk would be recommended.

Discussion

2. Addition of discussion of the challenges of obtaining such data in this population would add to the current discussion.  Given these potential barriers to data collection, is this likely to then have further inferences on the accuracy of data in this population (i.e. is it likely to overestimate or underestimate disease and why?)

3. In the discussion about designing care pathways, it may also be valid to highlight the need for collection of such data to be enabled within these systems in order to continue to get an accurate view of the oral health status of these populations and the impact on care.

4. Likewise, if we are targeting care towards older adults in LTC are we addressing the problem?  Certainly they may have improved access to care, but are we just managing disease or preventing disease?  This may be worthwhile discussing given your suggestion that dental issues may be exist at time of admission.  There is a lot of literature discussing this issue and why this may occur and it may be worthwhile discussing in this manuscript to ensure the reader does not assume this is a problem just associated with LTC.

Wishing you all the best for your ongoing research in this field.

Author Response

Thank you for taking the time to review our manuscript. I have numbered off your feedback points and placed our responses in italics below. In addition, changes within the manuscript are presented in yellow highlights.

  1. Introduction: While I think that the introduction sets a good background to the issue of interest, the caries process is oversimplified, particularly within the context of older adults in LTC.  Only biofilm and the influence of diet are discussed, even though these would not necessarily be any different from the rest of the population.  A more detailed discussion about other significant risk factors in this population that may place them at higher risk would be recommended.

 Thank you for pointing this out. The section has been updated to emphasise better the role of oral hygiene and the juxtaposition of the physiological state of older adults as follows:

“With regards to older adults, there are features such as a shift in their diet towards re-fined carbohydrates, complex medical conditions, polypharmacy, and diminishing dexterity for performing oral hygiene practices, which contribute to increases in the rate of oral disease activity. [4]”.

Discussion

  1. Addition of discussion of the challenges of obtaining such data in this population would add to the current discussion. Given these potential barriers to data collection, is this likely to then have further inferences on the accuracy of data in this population (i.e. is it likely to overestimate or underestimate disease and why?)

We have included comments on the inconsistency of data collection as it pertains to the disease indices used. We added comments on the challenges in sampling from the perspective of the sampling frame.

 “In addition, representative sampling is challenging due to difficulties ranging from logistical limitations associated with undergoing detailed dental examinations to gaining consent from those who might be cognitively impaired. These limitations mean that those with a more severe disease experience may be more likely to be excluded from sampling. As such, reports may underestimate the disease burden among this population.”

  1. In the discussion about designing care pathways, it may also be valid to highlight the need for collection of such data to be enabled within these systems to continue to get an accurate view of the oral health status of these populations and the impact on care.

Discussion, Para 6, sent 1 was updated as follows: “Designing a care pathway that caters to the needs of the older adults in the LTC population requires consistent and reliable data and along with that robust health surveillance systems”

  1. Likewise, if we are targeting care towards older adults in LTC are we addressing the problem? Certainly they may have improved access to care, but are we just managing disease or preventing disease? This may be worthwhile discussing given your suggestion that dental issues may be exist at time of admission.  There is a lot of literature discussing this issue and why this may occur and it may be worthwhile discussing in this manuscript to ensure the reader does not assume this is a problem just associated with LTC.

The authors agree that addressing oral disease in LTC, and only after they arrive in that setting is not the primary solution to addressing suffering as a result of oral disease among older adults. The following adjustment was made to highlight this concept: 

“The ideal approach would be to address oral disease at earlier stages of disease and while individuals possess greater levels of independence. In the context of LTC, the dental care delivery system is not well-positioned.”

Round 2

Reviewer 1 Report

Comments and Suggestions for Authors The manuscript has been clarified and I am satisfied with the responses to the comments.